# Identification of a Fatty Acid for Diagnosing Non-Alcoholic Steatohepatitis in Patients with Severe Obesity Undergoing Metabolic Surgery

**DOI:** 10.3390/biomedicines10112920

**Published:** 2022-11-14

**Authors:** Naoto Takahashi, Akira Sasaki, Akira Umemura, Tamotsu Sugai, Keisuke Kakisaka, Yasushi Ishigaki

**Affiliations:** 1Department of Surgery, Iwate Medical University School of Medicine, Morioka 028-3695, Japan; 2Division of Molecular Diagnostic Pathology, Department of Pathology, Iwate Medical University School of Medicine, Morioka 028-3695, Japan; 3Division of Hepatology, Department of Internal Medicine, Iwate Medical University School of Medicine, Morioka 028-3695, Japan; 4Division of Diabetes and Metabolism, Department of Internal Medicine, Iwate Medical University School of Medicine, Morioka 028-3695, Japan

**Keywords:** nonalcoholic steatohepatitis, laparoscopic sleeve gastrectomy, fatty acid, liquid chromatography-mass spectrometry, oxidized phospholipids, arachidonic acid cascade

## Abstract

The prevalence of nonalcoholic steatohepatitis (NASH) in severely obese Japanese patients is extremely high. However, there are currently no methods other than liver biopsy to assess hepatic steatosis and fibrosis. The purpose of this study was to comprehensively analyze changes in fatty acid (FA) and serum-free fatty acid (FFA) metabolism in severely obese Japanese patients to determine whether these could be surrogate markers. In this study, we enrolled 20 Japanese patients who underwent laparoscopic sleeve gastrectomy (LSG) for severe obesity and intraoperative liver biopsy. Serum FFAs were analyzed with liquid chromatography-mass spectrometry, and FAs in liver tissue were assessed using matrix-assisted laser desorption/ionization-imaging mass spectrometry to determine FAs that may be indicative of a positive NASH diagnosis. All patients showed significant weight loss and metabolic improvement following LSG. Regarding weight loss and metabolic improvement indices, 23 FFAs showed significant correlations with the baseline data. Narrowing down the phospholipids to commonly detected FAs detected in liver tissue, PC(18:1e_20:4) was significantly changed in the NASH group, suggesting that it could be used as a surrogate marker for NASH diagnosis. The results suggest that specific postoperative changes in blood phospholipids could be used as surrogate markers for NASH treatment.

## 1. Introduction

Nonalcoholic fatty liver disease (NAFLD) is the most common liver disease linked with metabolic syndrome, specifically obesity, which is a growing problem worldwide, as well as in Japan [1,2]. NAFLD subtypes include nonalcoholic fatty liver (NAFL) and nonalcoholic steatohepatitis (NASH), with NASH patients who suffer from advanced fibrosis being at increased risk of liver cirrhosis and liver disease-related death [2]. Metabolic surgery (MS), including laparoscopic sleeve gastrectomy (LSG), dramatically improved histological findings of NASH as well as type 2 diabetes (T2D); therefore, LSG and other MS may be strong alternatives for improving NASH [3,4].

Diagnosis of NASH is important for proper follow-up, including the prevention of disease progression and monitoring of liver function [5]. Currently, histopathological diagnosis is the only definitive way to diagnose NASH [6], but because of the risks associated with liver biopsy for severely obese patients, including bleeding and bile leakage, biomarkers and noninvasive methods are increasingly being used to screen patients at risk for NASH [7]. If effective biomarkers for NASH can be determined, the diagnosis of NASH can be made with less risk, and metabolic changes in liver tissue can be monitored over time.

Serum free fatty acid (FFA) has been reported to induce inflammatory responses and insulin resistance in obese patients [8,9,10], but very few reports have examined serum FFA changes after MS. Only a few papers have reported on the utility of FFA as a predictor of obesity and NASH, and the fatty acid (FA) species that have been examined thus far are limited [11]. Since NASH and severe obesity are closely related, we designed this study based on the hypothesis that lipid metabolism in the body and changes in serum FFA are somehow related to FA accumulation in liver tissues. The purpose of this study was to identify specific FAs as a potential marker for NASH diagnosis by evaluating associations between treatment response and FA in severely obese Japanese patients with and without NASH.

## 2. Materials and Methods

### 2.1. Patient Selection

This single-center retrospective study included data collection and analysis. Twenty severely obese patients who underwent LSG at Iwate Medical University Hospital between February 2020 and March 2021 were enrolled independently for this study. Patients were included if they met the insurance criteria for LSG along with the consensus statements on indication criteria for performing LSG for severely obese patients published in 2021 [12].

### 2.2. Data Collection

For all enrolled patients, clinical data, weight loss, and metabolic effects were evaluated at baseline and at 6 postoperative months (6POM). Weight, BMI, and % total weight loss (%TWL) were used as measures of the weight loss effect. Glucose metabolism-related parameters were fasting blood sugar (FBS), insulin, hemoglobin A1c (HbA1c), and C-peptide. The homeostasis model for the assessment of insulin resistance (HOMA-IR) was used for assessing insulin resistance, and the estimation of insulin secretory capacity was performed using the homeostasis model for assessing β-cell function (HOMA-β). NASH-related parameters included ferritin, type IV collagen 7S (T4C7S), aspartate aminotransferase (AST), and alanine aminotransferase (ALT). Visceral fat area (VFA) was measured by 64-row computed tomography (CT) (Toshiba Medical Systems Corporation, Tokyo, Japan) after acquiring 1 slice at umbilical height, in the Hounsfield unit (HU) range of −150 to 0 HU of adipose tissue and calculated as the area (cm^2^). In addition, liver volume (mL) and liver-to-spleen HU ratio (L/S ratio) were measured by CT using the imaging software SYNAPSE VINCENT (FUJIFILM, Tokyo, Japan). The NAFIC score [13], FIB-4 index [14], and NAFLD fibrosis score (NFS) [15] were calculated using baseline clinical data.

### 2.3. Lipidomics Analysis

Blood samples were collected after the 20 target patients had fasted for 8–10 h. Blood was separated by centrifugation (1800× *g*, 20 min) within 2 h after collection, and serum was frozen at −80 °C until analysis. Liver biopsies were also performed at the time of LSG, and the collected liver tissue was unfixed and frozen at −80 °C for storage. Regarding sample preparation, methanol to methyl tert-butyl ether and zirconia beads were added to the thawed serum and homogenized. Then, ultrapure water was added and centrifuged (3000× *g*, 10 min), and the methyl tert-butyl ether fraction was collected and filtered through a 0.2 μm filter. The same process was applied to an empty test to create a negative control sample. Negative control samples were used to confirm and remove the background noise introduced during sample preparation and liquid chromatography-mass spectrometry (LC-MS) analysis.

Each sample was analyzed using an Ultimate 3000 RSLC (Thermo Fisher Scientific K.K., Tokyo, Japan) HPLC system connected to a Q Exactive mass spectrometer (Thermo Fisher Scientific K.K., Tokyo, Japan). Separations were performed using a SunShell C18 column (ChromaNik Technologies Inc., Shizuoka, Japan). For each sample, ion peaks detected by LC-MS analysis, and retention time, intensity, exact mass, and MS spectral information were added. Based on the agreement of mass charge ratio (*m*/*z*) and retention time of all peaks detected in all samples, peaks commonly detected between samples (alignment members) were searched. Lipid molecular species were identified by comparing MS/MS spectra assigned to valid alignments according to the spectra of lipids in the Lipid Search (Kazusa DNA Research Institute, Tokyo, Japan) database [16]. From peak detection to alignment search, the data analysis software Power Get Batch (Kazusa DNA Research Institute, Tokyo, Japan) was used [17]. The alignments and the exact masses of known metabolites and theoretical compositional formulae were simultaneously searched in the UC2 database [18,19]. The search program MF Searcher (Kazusa DNA Research Institute, Tokyo, Japan) was used for the simultaneous database search [20]. Peak intensities were scaled using the Pareto method [21] so that the results of the principal component analysis could reflect the variation of peaks with weak intensities.

### 2.4. Matrix-Assisted Laser Desorption/Ionization-Imaging Mass Spectrometry (MALDI-IMS)

For liver tissue preparation, nine 1 mm diameter holes were made in a 1 cm^2^ area of frozen Tissue-Tek O.C.T. compound, and 8 patients’ liver tissues were buried unfixed; 1 hole was left empty and used as a negative control sample. After refreezing, the tissue was sliced into 10 μm thick slices in a Tissue-Tek Polar Cryostat/Microtome at −20 °C and mounted on the conductive surface of a conductive glass slide (ITO glass slide SIO100N, Matsunami Glass Industries, Osaka, Japan) for cryopreservation.

Sliced samples were deposited using an iM Layer (Shimadzu Corporation, Tokyo, Japan). IMS was performed in negative and positive ionization modes using an iMScope (Shimadzu Corporation, Tokyo, Japan) (Figure 1) [22]. Data were analyzed using the IMS Solution software package (Shimadzu Corporation, Tokyo, Japan). In this study, we focused our analysis on the phospholipids detected in the largest number of serum FFAs, which will be discussed below.

### 2.5. Liver Histology

All 20 severely obese patients enrolled in the study underwent intraoperative liver biopsies during LSG. Expert pathologists evaluated the diagnoses, the NAFLD activity scores (NAS), Brunt’s classification, and staging according to the evaluation methods proposed by the Japanese Society of Gastroenterology [23,24,25]. Based on the diagnosis of expert pathologists, patients diagnosed with NASH were classified into the NASH group, and those with other diagnoses (NAFL or normal liver) were classified into the non-NASH group. Ultrasound-guided liver biopsies were performed at 6POM in patients with intraoperative liver biopsies diagnosed as NASH. Patients who were still diagnosed with NASH at the 6POM liver biopsy were classified into the NASH continuation group, and those with other diagnoses were classified into the NASH improved group. We also evaluated the patients’ pericellular fibrosis score (PFS), which we originally devised [3].

### 2.6. Statistical Analysis

Data are presented as numbers and percentages for categorical variables and as means ± standard deviation for continuous variables. Statistical analysis was performed using the chi-squared test for categorical variables and Student’s *t*-test or the Mann-Whitney *u* test for continuous variables. For continuous variables, paired *t*-tests or Wilcoxon tests were used to allow for the comparison of all preoperative and postoperative parameters; a *p*-value < 0.05 was considered significant. For the creation of heatmaps, a *z*-Score was used to standardize and compare the data groups according to different units. For correlation analysis between serum FFA levels and various clinical data, Spearman’s rank coefficient (ρ) was calculated by fitting a straight line between the 2 variables. Receiver operating characteristic (ROC) curves were used to calculate cutoff values for each parameter. All statistical analyses were performed using JMP statistical software (version 14.2, SAS Institute, Cary, NC, USA).

## 3. Results

### 3.1. Patient Characteristics

The baseline mean weight was 118.7 kg, and BMI was 43.9 kg/m^2^. Nine patients had T2D, and intraoperative liver biopsy revealed the histopathologic diagnosis of NASH in 15 patients and non-NASH in 5 patients. Intraoperative liver biopsy histopathology was used to classify these patients into NASH and non-NASH groups (Table 1).

### 3.2. Weight Loss and Metabolic Improvement Effects

After LSG, mean body weight (118.7 kg vs. 89.8 kg; *p* < 0.001), VFA (252.6 cm^2^ vs. 146.4 cm^2^; *p* < 0.001), and liver volume (2259.1 mL vs. 1777.9 mL; *p* < 0.001) were significantly reduced, and %TWL was 23.6%. There were significant improvements in HOMA-IR (4.6 vs. 2.1; *p* < 0.001), significant decreases in AST (52.5 U/L vs. 20.4 U/L; *p* = 0.009), and ALT (69.7 U/L vs. 22.5 U/L; *p* = 0.002). There were significant reductions in body weight (120.4 kg vs. 91.6 kg; *p* < 0.001) and VFA (251.1 cm^2^ vs. 159.4 cm^2^; *p* = 0.003) in the NASH group and body weight (111.9 kg vs. 85.0 kg; *p* = 0.026) and VFA (259.7 cm^2^ vs. 128.7 cm^2^; *p* = 0.013) in the non-NASH group.

Comparing the histopathology findings between intraoperative and 6POM liver biopsies in the NASH group, liver tissue steatosis (25.7% vs. 7.7%; *p* = 0.001), NAS steatosis (1.3 vs. 0.6; *p* = 0.002), and NAS inflammation (0.5 vs. 0.1; *p* < 0.001) were significantly improved. For the non-NASH group, there was a mean liver tissue steatosis (30.6%) and a mean total NAS score (1.4), which consisted of 60% of the NAFL patients and 40% of the normal liver patients. Comparing the NASH improved and NASH continuation groups, only the PFS showed significant improvement (Table 2).

### 3.3. Serum FFA Analysis

Comprehensive analysis of serum FFA by LC-MS revealed 495 FAs. The numbers of saturated fatty acids (SFAs), mono-unsaturated fatty acids (MUFAs), and poly-unsaturated fatty acids (PUFAs) were 63, 102, and 330, respectively. When each lipid class was examined, phospholipids (275 species), ceramides (68 species), and sphingomyelins (67 species) accounted for 82.8% of all serum FFAs detected (Figure 2). The top 3 FFAs (phospholipids, ceramides, and sphingomyelin) with the highest number of detections were additionally examined, focusing on 189 FFAs that were commonly detected in all patients. A heat map of 189 FFAs was created based on the *z*-score of serum FFA levels and each patient’s background (Appendix A). From this heat map, it was difficult to compare the various fatty acids and each group.

FFAs that showed significant differences in levels in the baseline NASH and non-NASH groups, NASH, and non-NASH groups at 6POM, and in the NASH improved and NASH continuation groups at 6POM are shown in Table 3. Significant differences in level between the NASH and non-NASH groups were observed in 7 FFAs at baseline and in 4 FFAs at 6POM. No FFAs were found to change in conjunction with liver histology changes.

Postoperative changes in 189 FFA levels commonly detected in 20 patients are plotted in Figure 3, with the fold change on the horizontal axis and *p*-Values on the vertical axis as a volcano plot. There were 5 FFAs that significantly decreased postoperatively and 18 FFAs that significantly increased postoperatively. We generated a heatmap based on correlation coefficients for the weight loss index, diabetes index, liver enzymes, and NASH index (Figure 4). The heatmap showed a strong correlation of ρ = 0.8 or greater with the weight loss effect, glucose tolerance, and liver tissue steatosis in many FFAs.

Postoperative changes in 189 FFA levels commonly detected in 20 patients are plotted, with the fold change on the horizontal axis and *p*-Values on the vertical axis as a volcano plot.

We generated a heatmap based on correlation coefficients for the weight loss index, diabetes index, liver enzymes, and NASH index.

### 3.4. FA Analysis of Liver Tissue

MALDI-IMS analysis detected 22 phosphatidylcholines (PC), 16 phosphatidylinositols (PI), and 15 phosphatidylethanolamines (PE) in liver tissue. Of the 126 serum phospholipids detected by LC-MS, 87 phospholipids were extracted because their lipid class, carbon chain, and the number of unsaturated bonds were matched to the results of the MALDI-IMS analysis. A volcano plot was generated for that phospholipid (Figure 5), showing significant changes in three phospholipids, all of which increased postoperatively. In addition, a volcano plot of the NASH and non-NASH groups showed a significant postoperative increase in PC(18:1e_20:4) only in the NASH group (Figure 6). ROC curves were generated for serum PC(18:1e_20:4) levels, baseline BMI, HOMA-IR, and rate of liver steatosis, and a curve exceeding the area under the curve (AUC) 0.7 was calculated for baseline BMI (Figure 7).

It showed significant changes in three phospholipids, all of which increased postoperatively.

### 3.5. Validity as a Surrogate Marker

PC(18:1e_20:4) has oleic and arachidonic acids that are ether-linked, according to Lipid Search (Figure 8). A ROC curve was generated for the positive diagnosis of NASH in this FFA, with an AUC of 0.707, a cutoff of 20,711.3 ug/mL, and a positive diagnosis of NASH of 81.6%. NAFIC had a positive diagnostic rate of 75.0%, FIB-4 had a positive diagnostic rate of 6.7%, and NFS had a positive diagnostic rate of 73.3%, indicating that PC(18:1e_20:4) itself can be a surrogate marker for NASH in severely obese patients.

## 4. Discussion

In this study, we evaluated the patients with various NASH diagnostic scores that have been reported, but the correct diagnosis rate was low. Only PFS evaluated using liver tissue could detect significant differences between the NASH-improved and NASH continuation groups [3]. Thus, we hypothesized that variations in lipid metabolism and serum FFA levels in the body correlate with FA accumulation in liver tissue. However, although liver biopsies were performed for the diagnosis of NASH in previous reports, none were conducted with respect to FA changes in liver tissue. In this study, we used lipidomics analysis to investigate the changes in the composition and levels of serum FFA and FA in liver tissues before and after LSG in severely obese Japanese patients. We also determined FFA as a possible surrogate marker of LSG-induced NASH improvement.

In obese patients, nutrient intake exceeds consumption, causing body tissues to become saturated with lipids, leading to increased lipid transport and elevated serum FFA [26]. High levels of serum FFA are associated with insulin resistance through the inhibition of insulin-stimulated glucose uptake and glycogen synthesis [27]. Serum FFA levels are chronically elevated in obese patients [28] and investigating serum FFA levels has been proposed as a new approach to obesity treatment. Dietary fat is mainly composed of SFAs, MUFAs, and PUFAs [29]. The most common SFAs are palmitic and stearic acids, which are usually consumed in excess [30], and their increased intake leads to a higher risk of coronary artery disease and higher low-density lipoprotein cholesterol levels [31,32]. FAs accumulate in lipid droplets in adipocytes in the form of triglycerides, and excessive accumulation of FAs in systematic tissue leads to an increase in mature adipocyte size and exacerbates insulin resistance [33]. UFAs also include oleic acid, linoleic acid, and α-linolenic acid, which have been shown to have anti-inflammatory effects [34,35] and to directly modulate the activity of nuclear factor-kappa B (NF-κB), a major inflammatory transcription factor [36]. Furthermore, eicosatetraenoic acid and docosahexaenoic acid have established cardiovascular benefits [37]. Ni et al. performed Roux-en-Y gastric bypass as MS in obese patients and determined 17 SFAs, 10 MUFAs, and 13 PUFAs before and after surgery. The results showed little change in the level of SFAs, but the level of UFAs decreased significantly and progressively during the first 12 months postoperatively [38]. Previous studies have limited the types of FFAs detected. In this study, the comprehensive analysis of FFAs revealed many branched-chain UFAs in which SFAs and UFAs are combined. This may have contributed to the increase in the number of UFAs and the decrease in the SFA/UFA ratio.

Phospholipids are a major component of cell membranes and are also present in the blood as serum FFAs. They are susceptible to oxidation and readily produce oxidized phospholipids. Oxidized phospholipids have been shown to exhibit physiological activities, such as the regulation of apoptosis and inflammatory responses [39]. Phospholipids are increasingly being studied, as their oxidation may be associated with obesity, inflammation, and NASH. It has been reported that carbon chain (C)32, C34, and C36 are inversely related to the severity of NAFLD [40], that *n*-3 PUFA has anti-inflammatory effects in the liver [41], and that neutralizing oxidized phospholipids are effective treatment for NASH in studies in mice [42]. It has been reported that the increase in FFAs after LSG may be due to the mobilization of FFAs from adipose and hepatic tissues due to decreased dietary intake and decreased substrate availability for de novo lipid synthesis [7]. Similar changes occurred after LSG in this study, suggesting that FFA tended to increase after LGS.

We comprehensively examined the phospholipids detected in liver tissue that matched the type of serum phospholipids. PC(16:1e_18:1), PC(18:1_20:4), and PC(18:1e_20:4) showed significant increases in serum levels and a significant correlation with weight loss and metabolic improvement effects. Additional examination of serum phospholipid changes in the NASH group suggested that PC(18:1e_20:4) could be used as a surrogate marker for NASH diagnosis. PC(18:1e_20:4) has oleic acid (18:1) at the sn-1 position and arachidonic acid (20:4) at the sn-2 position, with acyl groups attached in ether form. These arachidonic acid-containing FAs are mobilized from glycerophospholipid membranes into the arachidonic cascade. The mobilized FAs release arachidonic acid by phospholipase A2, and inflammatory cytokines are produced from arachidonic acid by cyclooxygenase (COX) [43]. Without COX induction, reuptake into glycerophospholipids occurs via human acyl-CoA synthetase long-chain family member 4 [43]. Phospholipid levels, including PC(18:1e_20:4), increased after LSG. A possible explanatory mechanism for this effect could be that the progression of the arachidonic acid cascade was inhibited by the metabolic improvement caused by LSG, resulting in an increase in substrates prior to arachidonic acid production and increased reuptake of FFAs. Furthermore, it has been reported that FAs, such as phospholipids, may induce the activation of peroxisome proliferator-activated receptor alpha (PPAR-α), thereby inhibiting FA oxidation and contributing to decreased fat deposition in tissues [44]. Sustained induction of PPAR-α has been reported to suppress inflammation by inducing I kappa B kinase alpha, an inhibitory protein of NF-κB [44]. This effect may inhibit the progression of the arachidonic acid cascade and may also be involved in decreased inflammation in NASH. These effects may result in reduced inflammation throughout the body, especially in fatty liver tissue, and may contribute to the improvement of NASH in biopsy results.

There are several limitations to this study. First, the number of patients included was small because of the restricted analysis period and the expense involved in the lipidomics analysis. The methodology used in this study is very specialistic and obviously expensive; thus, it is surely not advisable for use to distinguish NASH from non-NASH in every epidemiological research, considering the high prevalence of these liver diseases. Second, the follow-up period was short. Longer-term studies involving a larger number of patients are needed before any final conclusions on this issue can be drawn. Third, lipidomics analysis could not be performed on all patients at the same time and had to be divided into three separate analyses for serum and liver tissue. The split analysis may have resulted in differences in the analysis methods and the levels of FA detected. Therefore, additional studies and validation cohorts are needed to eliminate these biases.

## 5. Conclusions

In conclusion, we demonstrated that there was an association between serum FFA and FA levels accumulated in liver tissues, and a correlation was also observed between changes in FFA levels and indices of weight loss and metabolic improvement. Phospholipids tended to increase after LSG, especially PC(18:1e_20:4) containing arachidonic acid, which was significantly increased only in the NASH group, and a ROC curve of AUC 0.717 was detected for the baseline BMI. LSG for severe obesity may inhibit the progression of the arachidonic acid cascade by suppressing the systemic inflammatory response through its metabolic improvement effect, which in turn may affect the improvement of NASH. In addition, PC(18:1e_20:4) could be used as a surrogate marker for NASH diagnosis and treatment response determination, as it significantly changed postoperatively only in the NASH group.

## Figures and Tables

**Figure 1 biomedicines-10-02920-f001:**
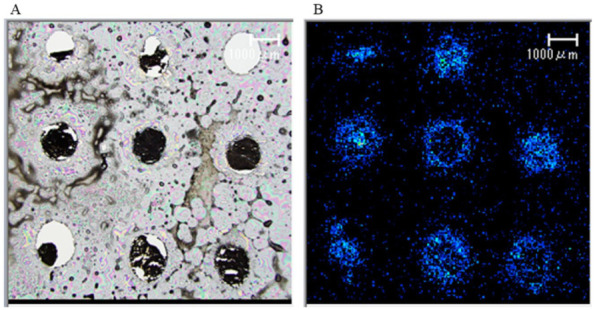
Optical microscopic and ionization spectral images of liver samples in MALDI-IMS. We made 9 1 mm diameter holes in a 1 cm^2^ area of frozen Tissue-Tek O.C.T. compound and 8 patients’ liver tissues buried unfixed; 1 hole was left empty and used as a negative control sample. The tissue was sliced into 10 μm thick slices and mounted on the conductive surface of a conductive glass slide. Sliced samples were analyzed by Matrix-assisted laser desorption/ionization-imaging mass spectrometry (MALDI-IMS). (**A**) Optical microscopic image of liver tissue. (**B**) Ionization spectral image of liver tissue.

**Figure 2 biomedicines-10-02920-f002:**
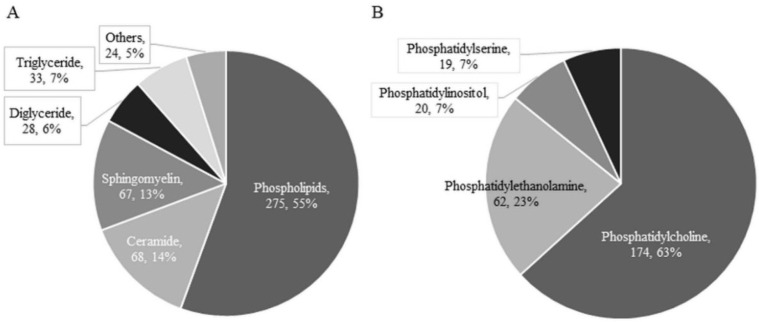
Breakdown of 495 serum FFA species detected by comprehensive analysis. (**A**) Breakdown of 495 FFAs. (**B**) Breakdown of 275 phospholipids.

**Figure 3 biomedicines-10-02920-f003:**
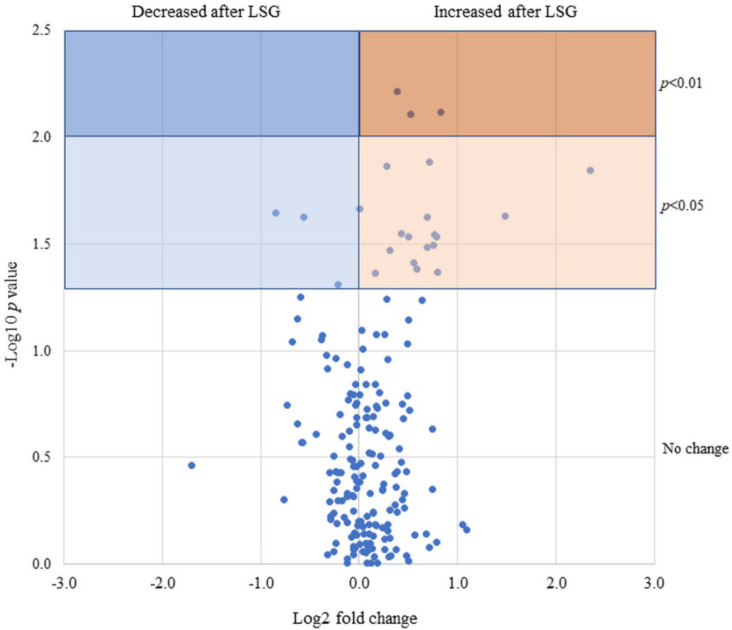
Volcano plot showing 189 FFAs based on fold change in serum levels after LSG.

**Figure 4 biomedicines-10-02920-f004:**
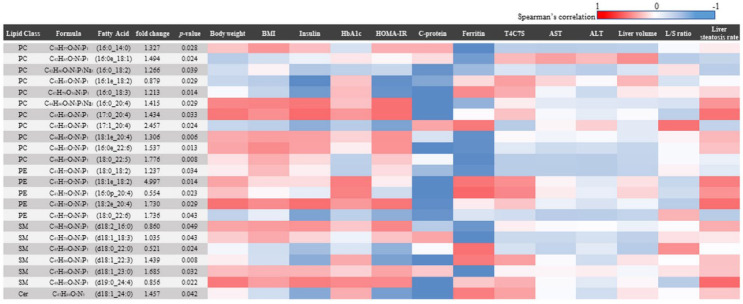
Correlation heatmap showing the relationship between FFA and various indices with significant serum level changes after LSG.

**Figure 5 biomedicines-10-02920-f005:**
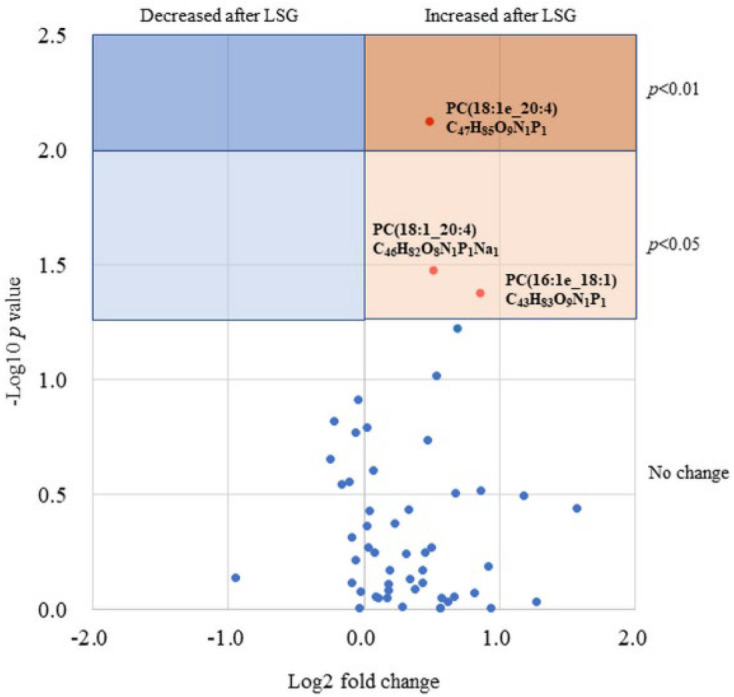
Postoperative changes in 53 phospholipids levels matched to their liver tissue. The blue-colored background means phospholipids with significant decrease, and the orange-colored background means phospholipids with significant increase between at baseline and after LSG.

**Figure 6 biomedicines-10-02920-f006:**
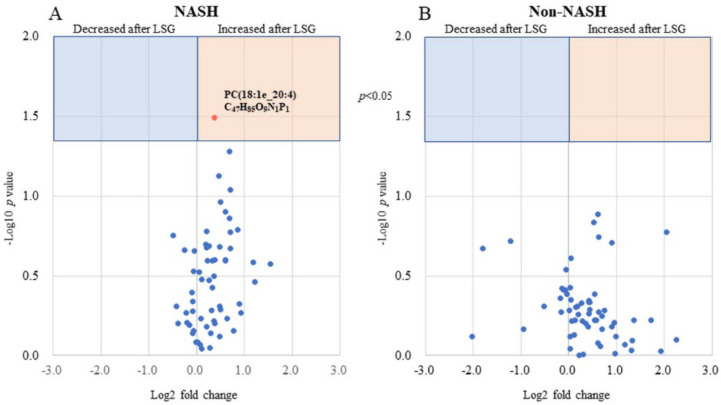
Differentiating between NASH and non-NASH groups, postoperative changes in phospholipids levels matched to their liver tissue. (**A**) A volcano plot of postoperative changes in phospholipids levels matched to their liver tissue in the NASH group. (**B**) A volcano plot of postoperative changes in phospholipids levels matched to their liver tissue in the non-NASH group.

**Figure 7 biomedicines-10-02920-f007:**
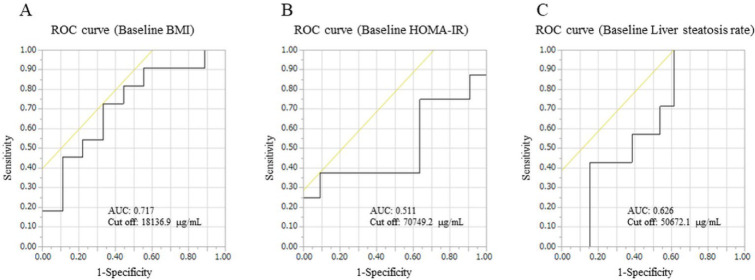
ROC curves were generated for serum PC(18:1e_20:4) levels with baseline parameters. (**A**) A ROC curve between PC(18:1e_20:4) and baseline BMI. (**B**) A ROC curve between PC(18:1e_20:4) and baseline HOMA-IR. (**C**) A ROC curve between PC(18:1e_20:4) and baseline liver steatosis rate. Black lines are ROC curves, and yellow lines means cut off values.

**Figure 8 biomedicines-10-02920-f008:**
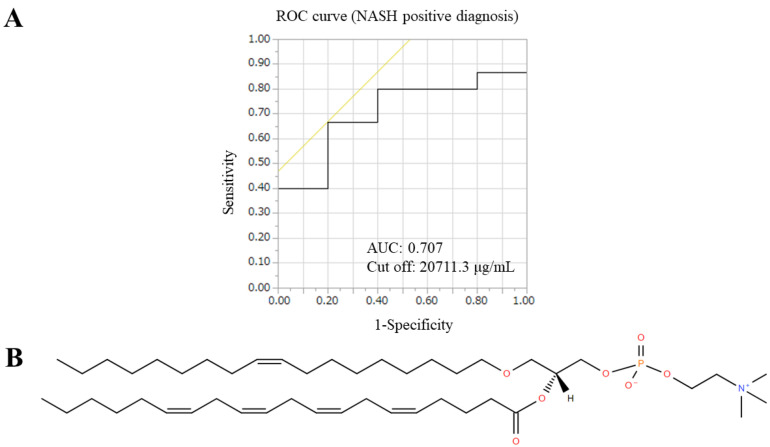
A ROC curve and a structural formula for determining the NASH-positive diagnostic rate in PC(18:1e_20:4). (**A**) A ROC curve for determining the NASH-positive diagnostic rate in PC(18:1e_20:4). (**B**) A structural formula of PC(18:1e_20:4).

**Table 1 biomedicines-10-02920-t001:** Baseline characteristics of enrolled patients.

	All Patients (*n* = 20)	NASH (*n* = 15)	Non-NASH (*n* = 5)	*p*-Value (NASH vs. Non-NASH)
Age (years)	38.3 ± 12.0	37.3 ± 12.3	42 ± 11.4	0.499
Male (*n*,%)	6, 30.0	5, 31.3	1, 25.0	0.819
T2D (*n*,%)	9, 45.0	8, 50.0	1, 25.0	0.395
Body weight (kg)	118.7 ± 21.8	120.4 ± 22.3	111.9 ± 21.5	0.500
BMI (kg/m^2^)	43.9 ± 4.9	44.8 ± 4.9	40.5 ± 2.9	0.115
Insulin (μU/mL)	16.8 ± 6.2	17.3 ± 6.7	14.9 ± 3.4	0.500
FBS (mg/dL)	114.4 ± 35.9	119.5 ± 38.2	94.0 ± 13.8	0.212
HbA1c (%)	6.8 ± 1.2	6.9 ± 1.2	6.0 ± 0.4	0.172
HOMA-IR (no unit)	4.6 ± 1.7	4.9 ± 1.8	3.4 ± 0.6	0.134
HOMA-β (no unit)	176.6 ± 126.7	168.9 ± 134.7	205.7 ± 101.5	0.620
C-peptide (ng/mL)	3.0 ± 1.2	3.1 ± 1.3	2.4 ± 0.2	0.465
Ferritin (ng/mL)	147.8 ± 178.8	162.6 ± 196.3	84.0 ± 33.0	0.511
T4C7S (ng/mL)	4.8 ± 1.1	4.9 ± 1.1	4.4 ± 1.0	0.452
VFA (cm^2^)	252.6 ± 87.9	251.1 ± 89.7	259.7 ± 97.3	0.884
Waist (cm)	120.8 ± 12.3	121.2 ± 13.1	118.7 ± 9.6	0.761
Liver volume (mL)	2259.1 ± 446.3	2337.6 ± 445.3	1945.3 ± 344.8	0.182
L/S ratio	0.8 ± 0.2	0.8 ± 0.2	0.9 ± 0.3	0.378
NAFIC (point)	1.6 ± 1.0	1.4 ± 0.9	2.0 ± 1.2	0.406
FIB-4 (point)	0.8 ± 0.8	0.9 ± 0.9	0.5 ± 0.2	0.100
NFS (point)	1.2 ± 1.7	1.5 ± 1.6	0.4 ± 2.0	0.303

Values are the mean ± standard deviation. Parameters with *p* < 0.05. Abbreviations. NASH, nonalcoholic steatohepatitis; T2D, type 2 diabetes; BMI, body mass index; FBS, fasting blood sugar; HOMA-IR, homeostasis model for the assessment of insulin resistance; HOMA-β, homeostasis model for assessing β-cell function; T4C7S, type IV collagen 7S; VFA, visceral fat area; L/S ratio, liver-to-spleen HU ratio; NFS, NAFLD fibrosis score.

**Table 2 biomedicines-10-02920-t002:** Liver Histopathology Comparison.

	NASH Baseline (*n* = 15)	NASH Continuation (*n* = 8)	NASH Improved (*n* = 7)	*p*-Value (Continuation vs. Improved)
Steatosis rate (%)	25.7 ± 17.1	7.1 ± 2.7	5.9 ± 6.4	0.638
PFS (point)	1.5 ± 0.9	2.1 ± 0.9	0.7 ± 0.8	0.010
NAS steatosis (point)	1.3 ± 0.5	0.8 ± 0.4	0.3 ± 0.5	0.094
NAS inflammation (point)	1.1 ± 0.4	0.6 ± 0.5	0.3 ± 0.5	0.317
NAS ballooning (point)	0.5 ± 0.6	0.3 ± 0.8	0.0 ± 0.0	0.356
NAS total (point)	2.9 ± 0.9	1.6 ± 1.4	0.6 ± 0.8	0.132
Brunt inflammation (point)	0.9 ± 0.3	1.1 ± 0.4	0.7 ± 0.5	0.093
Brunt fibrosis (point)	1.4 ± 0.6	1.3 ± 1.3	0.4 ± 0.8	0.156

Values are the mean ± standard deviation. Parameters with *p* < 0.05. Abbreviations. NAS, NAFLD activity score; PFS, pericellular fibrosis score; NASH, nonalcoholic steatohepatitis.

**Table 3 biomedicines-10-02920-t003:** List of FFAs with Significant Differences in Serum Levels in NASH vs. Non-NASH Groups.

		Baseline			6 Months after LSG			Continuation vs. Improved (6 Months after LSG)		
Lipid Class	Formula	NASH	Non-NASH	*p*-Value	NASH	Non-NASH	*p*-Value	NASH Continuation	NASH Improved	*p*-Value
PC(16:0e_16:1)	C_40_H_81_O_7_N_1_P_1_	7062.6	23,758.3	0.015	12,492.9	20,665.9	0.441	18,439.2	5697.1	0.239
PC(16:0e_18:1)	C_42_H_85_O_7_N_1_P_1_	4,569,271.9	11,999,844.0	0.047	5,198,314.3	12,383,645.0	0.098	6,060,208.6	4,213,292.3	0.654
PC(16:1e_18:2)	C_42_H_81_O_7_N_1_P_1_	298,943.3	770,318.8	0.050	315,576.7	605,506.6	0.030	395,094.5	224,699.2	0.164
PC(16:0e_20:4)	C_44_H_83_O_7_N_1_P_1_	7689.8	30,707.7	0.064	16,365.2	58,769.0	0.032	18,230.5	14,233.5	0.804
PC(18:1e_20:4)	C_46_H_85_O_7_N_1_P_1_	21,9594.4	277,865.5	0.224	344,802.5	436,341.1	0.050	401,461.4	280,049.5	0.419
PE(18:0_20:4)	C_43_H_77_O_8_N_1_P_1_	12,511.4	65,084.8	0.007	15,613.8	24,979.4	0.399	25,130.9	4737.1	0.069
PI(18:0_20:4)	C_47_H_82_O_13_N_0_P_1_	47,376.4	91,142.9	0.150	54,536.2	118,370.1	0.044	64,666.8	42,958.3	0.450
SM(d18:1_22:0)	C_46_H_92_O_8_N_2_P_1_	331,092.1	113,458.9	0.046	315,178.6	160,150.2	0.166	354,962.2	269,711.6	0.456
SM(d18:1_24:3)	C_47_H_90_O_6_N_2_P_1_	599,456.7	2,241,115.3	0.050	555,019.4	623,832.8	0.894	744,161.8	338,856.6	0.428
Cer(d18:1_23:0)	C_42_H_82_O_5_N_1_	95,317.8	442,889.9	0.040	127,706.1	185,562.2	0.473	150,156.7	102,048.2	0.558

Abbreviations. FFA, free fatty acid; LSG, laparoscopic sleeve gastrectomy; NASH, nonalcoholic steatohepatitis; PC, phosphatidylcholines; PE, phosphatidylethanolamines; PI, phosphatidylinositols; SM, sphingomyelin; Cer, ceramides.

## Data Availability

The data presented in this study are available on request from the corresponding author. The data are not publicly available due to the duty of confidentiality of all patients set by Japanese law; therefore, all data have been anonymized and strictly protected from external network connection.

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
