# Peer review of "Identification of a Fatty Acid for Diagnosing Non-Alcoholic Steatohepatitis in Patients with Severe Obesity Undergoing Metabolic Surgery"

_biomedicines, 2022, doi:10.3390/biomedicines10112920_

Round 1

Reviewer 1 Report

Dear author:

Please clarify how NASH patients were separated into NASH-continuous and NASH-improved groups after LSG.

According to Table 1, BMI, NAFIC score, FIB-4 score, and NFS score were not significantly different between Non-NASH and NASH patients. As a better surrogate marker, does the level of PC(18:1e_20:4) significantly different between Non-NASH and NASH patients before the surgery?

Author Response

6 November, 2022

Editors

Biomedicines

Editors:

We are grateful that you are willing to reconsider our manuscript, and we sincerely appreciate the valuable and constructive comments of the reviewers. We believe that these comments have significantly improved the quality of our manuscript. To address the reviewers’ concerns, we have substantially revised the original manuscript. Our point-by-point responses to the reviewers’ comments are provided below.

Reviewer #1:

  1. Please clarify how NASH patients were separated into NASH-continuous and NASH-improved groups after LSG.

Thank you for your reasonable suggestion. We modified the paragraph of liver histology along with your indication. L153-159, “Based on the diagnosis of expert pathologists, patients diagnosed with NASH were classified into the NASH group and those with other diagnoses (NAFL or normal liver) were classified as the non-NASH group. Ultrasound-guided liver biopsies were performed at 6POM in patients with intraoperative liver biopsies diagnosed as NASH. Patients who were still diagnosed with NASH at the 6POM liver biopsy were classified into the NASH continuation group, and those with other diagnoses were classified into the NASH im-proved group. And we also evaluated the patients’ pericellular fibrosis score (PFS), which we originally devised [15]”.

  1. According to Table 1, BMI, NAFIC score, FIB-4 score, and NFS score were not significantly different between Non-NASH and NASH patients. As a better surrogate marker, does the level of PC(18:1e_20:4) significantly different between Non-NASH and NASH patients before the surgery?

We agree with your assessment. We have added the level of PC(18:1e_20:4) in Table 3, and modified the applicable sentence; L221-223, “Significant differences in level between the NASH and non-NASH groups were observed in 7 FFAs at baseline and in 4 FFAs at 6POM”.

Sincerely,

Akira Sasaki, M.D., Ph.D.

Department of Surgery, Iwate Medical University School of Medicine

2-1-1 Idaidori, Yahaba, 028-3695, Japan.

Reviewer 2 Report

Dear authors,

I read with interest the paper entitled “Identification of a fatty acid for diagnosing non-alcoholic steatohepatitis in patients with severe obesity undergoing metabolic surgery”.

I found the topic of this paper interesting.

Although the aim of the study is original and important, there are several issues that the authors should address to improve the manuscript:

1.     The introduction section should be shorter and in parts rewritten. Chose the term you want to use – NAFLD or MAFLD based on the criteria you used in your study. MAFLD has not entered the official guideline. The statement that all obese patients are automatically diagnosed with MAFLD should be rephrased.

2.     The aim of this study is not clear. Please rewrite. The main aim is to define changes in FAs in NASH vs NAFLD before and after LSG? The current knowledge gaps in FAs in NAFLD should be described in the introduction.

3.     Results: Have these patients already been reported in previous publications?  Serum FFA analysis – Figure 3 is as you mentioned difficult to read – perhaps to include it in Supp. Materials instead of the main manuscript? Limitation of statistical methods – the question of the validity of ROC analysis since you had only 15 patients with NASH and 5 NAFL?

4. The discussion is too long and must be completely rewritten. The discussion section is not a review of the literature. It is reasonable that in the first paragraph the authors highlight their main findings or introduce the reader to the discussion with a short paragraph in the context of the proposed issue. Afterward, compare with previous results in the literature. Point out the similarities and differences between your work and the manuscripts analyzed, discuss the probable mechanisms, and make a hypothesis of the obtained results.

Author Response

6 November, 2022

Editors

Biomedicines

Editors:

We are grateful that you are willing to reconsider our manuscript, and we sincerely appreciate the valuable and constructive comments of the reviewers. We believe that these comments have significantly improved the quality of our manuscript. To address the reviewers’ concerns, we have substantially revised the original manuscript. Our point-by-point responses to the reviewers’ comments are provided below.

Reviewer #2:

I read with interest the paper entitled “Identification of a fatty acid for diagnosing non-alcoholic steatohepatitis in patients with severe obesity undergoing metabolic surgery”.

I found the topic of this paper interesting.

Although the aim of the study is original and important, there are several issues that the authors should address to improve the manuscript:

  1. The introduction section should be shorter and in parts rewritten. Chose the term you want to use – NAFLD or MAFLD based on the criteria you used in your study. MAFLD has not entered the official guideline. The statement that all obese patients are automatically diagnosed with MAFLD should be rephrased.

Thank you for your suggestion. We have modified our introduction without using the concept of MAFLD. The following sentences have been inserted; L47-50, “However, there are some cases that do not respond well to weight loss and metabolic improvement through pharmacotherapy and lifestyle modifications. For such cases, metabolic surgery (MS) is known to contribute to effective weight loss and the improvement of obesity-related diseases [11, 12]”.

  1. The aim of this study is not clear. Please rewrite. The main aim is to define changes in FAs in NASH vs NAFLD before and after LSG? The current knowledge gaps in FAs in NAFLD should be described in the introduction.

Thank you for your kind suggestion. We modified the aim of this study in introduction; L73-76, “The purpose of this study was to identify specific FAs as a potential marker for NASH diagnosis by evaluating associations between treatment response and FA in Japanese severely obese patients with and without NASH”.

In addition, current knowledge gaps in FAs in NAFLD/NASH have not been clarified well. The following sentence have been also added in the introduction section; L66-68, “Only a few papers have reported on the utility of FFA as a predictor of obesity and NASH, and the fatty acid (FA) species that have been examined thus far are limited [23]”.

  1. Results: Have these patients already been reported in previous publications?

These patients were independently enrolled in this study. L76-78, “This single-center retrospective study included data collection and analysis. Twenty severely obese patients who underwent LSG at Iwate Medical University Hospital between February 2020 and March 2021 were enrolled independently for this study”.

  1. Serum FFA analysis – Figure 3 is as you mentioned difficult to read – perhaps to include it in Supp. Materials instead of the main manuscript?

Thank you for the concise suggestion. We moved these figures to supplementary materials with their legends.

  1. Limitation of statistical methods – the question of the validity of ROC analysis since you had only 15 patients with NASH and 5 NAFL?

Thank you for providing these insights. The ROC curves were calculated by 20 patients enrolled in this study. The other reviewer also indicated that the methodology of this study was very specialistic and obviously expensive; therefore, further additional studies might be warranted with gathering larger number of patients in near future. By these social issues, we could not enroll validation cohort analyses in this study. We described about these limitations in L360-370.

  1. The discussion is too long and must be completely rewritten. The discussion section is not a review of the literature. It is reasonable that in the first paragraph the authors highlight their main findings or introduce the reader to the discussion with a short paragraph in the context of the proposed issue. Afterward, compare with previous results in the literature. Point out the similarities and differences between your work and the manuscripts analyzed, discuss the probable mechanisms, and make a hypothesis of the obtained results.

We have reflected this comment by omitting some sentences in the discussion section.

Sincerely,

Akira Sasaki, M.D., Ph.D.

Department of Surgery, Iwate Medical University School of Medicine

2-1-1 Idaidori, Yahaba, 028-3695, Japan.

Reviewer 3 Report

Authors in the Introduction section, dealing with the therapeutical approaches for NASH, state that......However, the long-term maintenance of weight loss  through drug therapy and lifestyle modifications is difficult to achieve. In contrast, metabolic surgery  is known to contribute to effective weight loss and the improvement of obesity-related diseases...this is correct, but they should honestly recognise that there are many drugs that are reckoned as good candidates to cure NASH, as evident in various recent papers, for example...  Insights into the molecular targets and emerging pharmacotherapeutic interventions for nonalcoholic fatty liver disease. Metabolism. 2022 Jan;126:154925. doi: 10.1016/j.metabol.2021.154925. Epub 2021 Nov 2. PMID: 34740573.

Authors should better pointed out the characteristics at histology of the five patients belonging to the  non-NASH cohort, in order to evidence the differences between the two groups.

Authors should emphasise in the Limitations section that the methodology proposed in this study is very specialistic and obviously expensive, thus surely is not advisable to be used to distinguish NASH from non-NASH in every epidemiological research, considering the high prevalence of these liver diseases.

Author Response

6 November, 2022

Editors

Biomedicines

Editors:

We are grateful that you are willing to reconsider our manuscript, and we sincerely appreciate the valuable and constructive comments of the reviewers. We believe that these comments have significantly improved the quality of our manuscript. To address the reviewers’ concerns, we have substantially revised the original manuscript. Our point-by-point responses to the reviewers’ comments are provided below.

Reviewer #3:

  1. Authors in the Introduction section, dealing with the therapeutical approaches for NASH, state that......However, the long-term maintenance of weight loss through drug therapy and lifestyle modifications is difficult to achieve. In contrast, metabolic surgery is known to contribute to effective weight loss and the improvement of obesity-related diseases...this is correct, but they should honestly recognize that there are many drugs that are reckoned as good candidates to cure NASH, as evident in various recent papers, for example... Insights into the molecular targets and emerging pharmacotherapeutic interventions for nonalcoholic fatty liver disease. Metabolism. 2022 Jan;126:154925. doi: 10.1016/j.metabol.2021.154925. Epub 2021 Nov 2. PMID: 34740573.

Thank you very much for your indications. We cited the indicated paper as Ref#10 in introduction, and we have also modified the following sentences; L47-50, “However, there are some cases that do not respond well to weight loss and metabolic improvement through pharmacotherapy and lifestyle modifications. For such cases, metabolic surgery (MS) is known to contribute to effective weight loss and the improvement of obesity-related diseases [11, 12]”. And, also following sentences have been inserted into the introduction section; L56-63, “Diagnosis of NASH is important for proper follow-up, including the prevention of disease progression, and monitoring of liver function [17]. Currently, histopathological diagnosis is the only definitive way to diagnose NASH [18], but because of the risks associated with liver biopsy for severely obese patients including bleeding and bile leakage, biomarkers and noninvasive methods are increasingly being used to screen patients at risk for NASH [19]. If effective biomarkers for NASH can be determined, the diagnosis of NASH can be made with less risk, and metabolic changes in liver tis-sue can be monitored over time”.

  1. Authors should better point out the characteristics at histology of the five patients belonging to the non-NASH cohort, in order to evidence the differences between the two groups.

We agree with you and have incorporated this suggestion throughout our paper. The following sentence has been inserted in the result section; L198-200, “For the non-NASH group, there was a mean liver tissue steatosis (30.6%) and a mean total NAS score (1.4), which consisted of 60% NAFL patients and 40% normal liver patients”.

  1. Authors should emphasize in the Limitations section that the methodology proposed in this study is very specialistic and obviously expensive, thus surely is not advisable to be used to distinguish NASH from non-NASH in every epidemiological research, considering the high prevalence of these liver diseases.

Thank you for your suggestion. We have inserted the sentences about your opinion in the discussion section; L362-364, “The methodology used in this study is very specialistic and obviously expensive, thus surely is not advisable to be used to distinguish NASH from non-NASH in every epidemiological research, considering the high prevalence of these liver diseases”.

Sincerely,

Akira Sasaki, M.D., Ph.D.

Department of Surgery, Iwate Medical University School of Medicine

2-1-1 Idaidori, Yahaba, 028-3695, Japan.

Round 2

Reviewer 2 Report

The authors have significantly improved the manuscript. 

However, I believe that the Introduction and Discussion are still too long. 

Lines 34-55 are not necessary - just describe in few sentences that NAFLD is the most common CLD, linked with metabolic syndrome, specifically obesity which is growing problem worldwide as well as in Japan. 

Some parts of discussion can be shorter - e.g. lines 311-314 can be deleted

328-331 - you do not need to repeat the results in the discussion. 

Author Response

November 10, 2022

Dr. Shaker A. Mousa

Editor-in-chief, Biomedicine

Dear. Dr. Shaker A. Mousa

The comments of the three reviewers have been helpful in allowing us to our manuscript. We have attempted to address the questions raised by the reviewers according to the following:

Reviewer #2:

The authors have significantly improved the manuscript.

However, I believe that the Introduction and Discussion are still too long.

Lines 34-55 are not necessary - just describe in few sentences that NAFLD is the most common CLD, linked with metabolic syndrome, specifically obesity which is growing problem worldwide as well as in Japan.

Thank you for your concise suggestion. We shortened the relevant introduction as follows; L34-41, “Nonalcoholic fatty liver disease (NAFLD) is the most common liver disease, linked with metabolic syndrome, specifically obesity which is growing problem worldwide as well as in Japan [1,2]. NAFLD subtypes include nonalcoholic fatty liver (NAFL) and nonalcoholic steatohepatitis (NASH), with NASH patients who suffer from advanced fibrosis being at increased risk of liver cirrhosis and liver disease-related death [2]. Metabolic surgery (MS) including laparoscopic sleeve gastrectomy (LSG) dramatically improved histological findings of NASH as well as type 2 diabetes (T2D); therefore, LSG and other MS may be strong alternatives for improving NASH [3, 4]”.

Some parts of discussion can be shorter - e.g. lines 311-314 can be deleted

We have reflected this comment by omitting the relevant sentences.

328-331 - you do not need to repeat the results in the discussion.

Thank you for your suggestion. We have reflected this comment by omitting some sentences in the discussion section.

Sincerely,

Akira Sasaki, M.D., Ph.D.

Department of Surgery, Iwate Medical University School of Medicine

2-1-1 Idaidori, Yahaba, 028-3695, Japan.